# Fe_3_O_4_-Nanoparticle-Doped Epoxy Resin as a Detachable Adhesive by Electromagnetic Heating for GFRP Single-Lap Joints

**DOI:** 10.3390/nano12213913

**Published:** 2022-11-06

**Authors:** Xoan F. Sánchez-Romate, Antonio del Bosque, Anabel Crespo, Rafael Alonso, María Sánchez, Alejandro Ureña

**Affiliations:** 1Materials Science and Engineering Area, Escuela Superior de Ciencias Experimentales y Tecnología, Universidad Rey Juan Carlos, Calle Tulipán s/n, 28933 Móstoles, Spain; 2AIMPLAS Instituto Tecnológico del Plástico, Carrer de Gustave Eiffel 4, 46980 Paterna, Spain

**Keywords:** adhesive joints, GFRP, electromagnetic heating, single-lap shear, composite structures

## Abstract

An adhesive based on a Fe_3_O_4_-nanoparticle (MNP)-doped epoxy resin was proposed for the development of detachable adhesive joints with GFRP substrates. The analysis of cryofractures showed that the increasing MNP content promotes a higher presence of larger aggregates and a lower sedimentation of nanoparticles due to the higher viscosity of the mixture. In this regard, the inclusion of expandable microspheres (MS) induces a more uniform dispersion of MNPs, reducing their sedimentation. The capability of the proposed adhesives for electromagnetic (EM) heating was also evaluated, with increases in temperature of around 100 °C at 750 A, enough to reach the T_g_ of the polymer required to facilitate the adhesive detachment, which is around 80 °C. Finally, the lap shear strength (LSS) of 14 and 20 wt.% MNP samples was evaluated in a single-lap shear joint with simultaneous EM heating. The LSS values were reduced by 60–80% at 750 A, thus promoting successful adhesive joint detachment under EM heating.

## 1. Introduction

Nowadays, there is an increasing tendency towards the use of complex structures for a wide range of applications. For this reason, it is necessary to develop proper assembly techniques that guarantee the reliability of these structures.

In this regard, the development of structural adhesives is gaining a great deal of attention [1]. More specifically, adhesive joints present a more uniform stress distribution, help to avoid possible corrosion issues, and promote weight saving in comparison to mechanical joints [2,3]. In fact, recent studies have dealt with the development of novel joining techniques, such as electromagnetic-driven, self-piecing riveting for joints with dissimilar materials and with Al/steel substrates, examining their mechanical performance and corrosion properties [4], as well the joining quality [5] in comparison to adhesive and hybrid joints.

However, adhesives also present some important limitations concerning, in particular, their reliability, making them very difficult to inspect. Therefore, there is great interest in developing proper inspection techniques that offer detailed information about the health of these joints. For example, the use of low-cost techniques based on electrical impedance measurements has been proved to be an effective way of detecting the failure mode [6]. On the other hand, the use of carbon nanotube-doped adhesive films has demonstrated a good efficiency for detecting fatigue crack propagation [7] and flaws under quasi-static conditions in skin-stringer elements [8] or in Mode-I standard specimens [9].

Another important limitation of adhesive joints is that they are permanent, which means that their repair and recycling are quite difficult, for example. In this context, the development of removable or detachable adhesive joints is gaining a great deal of interest. 

The detachment of adhesive joints is usually based on the destruction of the joint via thermal or chemical degradation, but these methods may often degrade the properties of the substrates at the general detriment of the mechanical performance of the overall structure. For this reason, there is increasing interest in the development of detachable or reversible adhesives. In fact, several studies have studied the requirements that detachable adhesives must accomplish in the manufacturing process, with adhesive formulation and the type of substrate being the most important parameters to be considered [10,11].

There are several methods used to ensure the proper detachment of the adhesive joint. These include the addition of microcapsules with polar dissolvents, including additives, to facilitate the detachment process. For example, the use of microcapsules of amines and metallic halides has been reported as an effective method for promoting the excision of an epoxy matrix [12].

Furthermore, the addition of chemical or physical foaming agents (CFA, PFA) has been also explored as an effective way of promoting the adhesive detachment. They are based in a volumetric expansion due to the influence of a chemical (CFA) or a physical agent (PFA), typically the temperature [13,14,15]. 

In this regard, the use of thermally expanded particles (TEPs) has been widely investigated for the development of this type of adhesive. More specifically, the effect of thermally expandable microspheres led to a drastic reduction in the joint strength at 80–100 °C due to the volumetric expansion of the microspheres at these temperatures [16,17,18].

This thermal activation can be achieved by means of external heating, for example, in a conventional oven, or via other sources. For example, the process of heating activation by Joule’s effect has been widely explored. Here, the heat, *Q*, is estimated from the voltage applied, *V*, the current passing through the material, *I*, and the time, *t*, and the electric field is applied, following this formula: (1)Q=V·I·t

To achieve heating activation by Joule’s effect, it is thus necessary to have a network that responds to the application of an electric field. 

A common method used to achieve this effect in polymeric materials is the addition of electrically conductive nanoparticles. More specifically, it has been widely proved that the addition of small numbers of carbon nanotubes (CNTs) and graphene nanoplatelets (GNPs) promotes a temperature increase by Joule’s effect of more than 100 °C at relatively low voltages, demonstrating the efficiency of this type of heating [19,20,21,22]. Furthermore, the Joule’s heating capabilities have been used for a wide range of purposes, such as the development of de-icing systems [23,24] and thermally activated self-healable systems [25,26], or resistance welding in thermoplastic composites [27].

In this regard, electromagnetic (EM) heating is also gaining a great deal of attention. The basis of this technique is quite similar to that of electrothermal activation and lies in the fact that conductive micro- and nanoparticles embedded in a polymeric matrix can act as heaters when exposed to an electromagnetic field. More specifically, the EM field induces eddy currents that promote the temperature increase via Joule’s heating. Through electrothermal activation, the EM heating enables targeted heating, as well as less energy consumption when compared to conventional heating. For this reason, it can be used for a wide range of applications, such as the welding of thermoplastic composites [28] or acerated curing of adhesives [29,30], among others, such as biomedical purposes [31].

The use of ferromagnetic nanoparticles such as Fe_3_O_4_ has been widely explored to promote the thermal degradation of thermoplastic adhesives via EM heating, facilitating the detachment process. More specifically, when used to induce thermal degradation in ABS [32,33] and polypropylene copolymers [34,35], they proved to have good heating capabilities in comparison to other sources, such as microwave heating. However, their use in thermosetting adhesives for structural joints remains to be explored. 

Therefore, this work aims to investigate the development of detachable adhesive joints based on EM heating via the inclusion of Fe_3_O_4_ nanoparticles in a thermosetting adhesive. To achieve this purpose, the nanoparticles were dispersed under ultrasonication, and a process optimization was carried out. Then, the temperature increase as a function of nanoparticle content was studied, and the effect of the inclusion of the expandable microspheres was also investigated. Finally, a proof-of-concept of the joint detachment through a single-lap shear test was conducted to prove the efficiency of the detachment process in the proposed adhesives. Here, the main novelty of this work is the detailed monitoring of the lap shear strength value of the joints as a function of the MNP content and induction current in a thermosetting adhesive. 

## 2. Materials and Methods

### 2.1. Materials

The adhesive is based on a mixture of epoxy resin, magnetic nanoparticles (MNPs), and microspheres (MS). In order to compare the effects of each component, different compositions were manufactured: neat epoxy resin, MNP-reinforced epoxy resin, and MNP/MS-reinforced epoxy resin.

Epoxy resin was purchased from Resoltech (Resoltech, Rousset, France) as a commercial formula called 1070ECO. The formulation incorporates a 37 wt.% bio-content in its reactive diluent, without compromising its mechanical properties. The monomer has a viscosity of 1750 mPa·s, and the mixture between the monomer and the hardener has a viscosity of 700 mPa·s at 23 °C. 

The Fe_3_O_4_ magnetic nanoparticles (MNPs) with a spherical nano-powder shape were supplied by Sigma-Aldrich (SigmaAldrich, St. Louis, MO, USA). They have a 97% purity, an average particle size of 50–100 nm, and a BET surface area of 6–8 m^2^/g.

The thermoplastic-expanded microspheres (MS) were supplied by Expancel DU (Nouryon, Amsterdam, The Netherlands) with the commercial name 031DU40. They have a density of 12 kg/m^3^ and an unexpanded particle size of 10–16 μm, measured by low-angle laser light scattering. They start to expand at 80–95 °C, with an expanded particle size of around 40 μm. 

### 2.2. Manufacturing of MNP/Epoxy Nanocomposites and Adhesive Joints

The nanocomposites were prepared as follows: First, the nanoparticles (MNP or MNP/MS) were dispersed in the epoxy monomer by an ultrasonication process using a HIELSCHER ULTRASONIC PROCESSOR UP400ST (Hielscher Ultrasonics, Teltow, Germany) machine at a 50% amplitude with 0.5 pulses. Ultrasonication induces the breakage of nanoparticle agglomerates owing to the cavitation forces induced by the ultrasonic pulses. After the ultrasonication process, the mixture was heated to 80 °C and degassed for 20 min to remove possible entrapped air. In this way, the use of magnetic stirrers can be avoided due to the ferromagnetic nature of the nanoparticles. Then, the epoxy hardener was added in 100:35 mass proportions (monomer:hardener) and manually mixed. At this point, there were differences in the manufacturing of the bulk nanocomposites and SLS specimens.

One the one hand, for the bulk nanocomposites, the mentioned mixture was poured into a metallic mold previously smeared with a layer of release agent supplied by Castro Composites (Castro Composites, Porriño, Spain), and they were cured in an oven at 60 °C for 16 h, following the indications provided by the supplier. The compositions of all the bulk nanocomposites prepared and their nomenclature are listed in Table 1.

One the other hand, the single-lap shear (SLS) samples were prepared according to ASTM D 5868-95, using GFRP substrates of 100 × 25.4 × 2.5 mm with an overlapping area of 25.4 × 25.4 mm and 1 mm thickness of the nanocomposite adhesive, following the same curing cycle as that mentioned above. The adhesive thickness was controlled using metallic tabs during the fabrication. In this way, it was possible to adjust the adhesive thickness during the curing cycle. In this regard, only the 14 and 20 wt.% MNP samples (with and without MS) were manufactured, because the lower MNP contents did not reach a temperature high enough to promote the proper detachment of the adhesive by EM heating.

### 2.3. Characterization of Nanocomposites

#### 2.3.1. Microstructural Characterization

The nanoparticles’ dispersion in the MNP/epoxy mixtures was analyzed by light-transmitted optical microscopy (TOM) prior to the addition of the hardener, that is, before the curing step. A study of the influence of the sonication time on the dispersion state was carried out, taking samples after 15 min, 30 min, 1 h, 2 h, and 3 h of sonication. The microscope used was a Leica DMR (Leica Microsystems, Wetzlar, Germany) equipped with a camera, the Nikon Coolpix 990 (Nikon, Tokyo, Japan).

Furthermore, in order to evaluate the dispersion state obtained by these nanocomposites after the curing process, the analysis of the fracture surfaces under cryogenic conditions was performed by scanning electron microscopy (SEM) using an S–3400N machine from Hitachi. The nanocomposite cryofractures were coated by a thin layer of gold to achieve electrically conductive surfaces.

#### 2.3.2. Thermomechanical Characterization 

The thermomechanical properties were studied by dynamic mechanical thermal analysis (DMTA) using a TA DMTA Q800 (TA Instruments, New Castle, DE, USA) machine, according to standard ASTM 5418. At least two samples with dimensions of 35 × 12 × 1.25 mm for each condition listed in Table 1 were tested in the single cantilever mode over a temperature range from 20 to 160 °C, with a heating ramp of 2 °C/min and 1 Hz frequency. The glass transition temperature (T_g_) was measured as the maximum of the tanδ curve, while the storage modulus (E′) was determined at 25 °C for every sample. 

#### 2.3.3. Electromagnetic Heating Characterization 

The EM heating capabilities were analyzed by applying an external magnetic induction to a sample of 10 × 10 × 1 mm using a VEVOR High-Frequency Induction Heater Furnace (VEVOR, Rancho Cucamonga, CA, USA) machine, with a power of 15 kW and a working frequency range of 30–100 kHz. In this regard, the sample was placed inside a 7 cm-diameter solenoid, which applied the EM field. The temperature reached by the nanocomposite was measured with an infrared thermographic camera FLIR E50 (Teledyne FLIR, Wilsonville, OR, USA). The induction was applied for 3–5 min to guarantee the stabilization of the temperature of the sample. For each nanocomposite, temperature measurements were obtained for the current intensities of the induction furnace solenoid between 250 and 750 A in steps of 100 A.

### 2.4. Single-Lap Shear Tests 

With the aim of proving the applicability of the nanocomposites as detachable adhesives, SLS tests were carried out by the simultaneous application of EM heating. Thus, at least two specimens for each condition were tested to analyze the effects of the MNP and MNP/MS contents on the mechanical properties and detachability of the adhesive joints. SLS test were conducted according to ASTM D 5868-95 using an MTS Alliance RF/100 (MTS Systems Corporation, Eden Prairie, MN, USA) test machine with a load cell of 30 kN at a test rate of 13 mm/min. Before and simultaneously with this mechanical test, the adhesive joints were heated by induction with a solenoid at 250, 500, and 750 A for 4 min to reach the stabilization temperature. The experimental setup of the proof-of-concept is shown in Figure 1.

## 3. Results

In this section, an analysis of the dispersion procedure is carried out, as well as the microstructural characterization of the fracture surfaces of the nanocomposites. Then, their mechanical properties under DMTA testing and their heating capabilities based on induction are evaluated. Finally, a proof-of-concept of the detachable adhesives under the effect of EM heating is offered based on single-lap shear tests. 

### 3.1. Microstructural Analysis

Figure 2 shows several TOM images of the dispersion of the MNPs at different sonication times to evaluate the optimum processing conditions. In this regard, it can be observed that, at low sonication times, there is a high prevalence of aggregates in the mixture (Figure 2a,b). By increasing the sonication time, the efficiency of the cavitation forces induced by ultrasonication is increased [36], leading to a reduction in the number of larger aggregates (Figure 2c,d). At the greatest sonication time measured, that is, 3 h, the dispersion state achieved in the mixture does not present any prevalent larger aggregates (Figure 2e). In addition, due to the spheric geometry of the MNP, it was not expected to show any prevalent damage in the nanoparticles. These statements are confirmed by observing the histogram in Figure 2f. Here, it is observed that the percentage of small size particles (less than 2 µm) is increased with the sonication time, indicating that there is a significant breakage of aggregates due to the higher efficiency of the sonication process. Conversely, a reduction in the aggregates of larger size (>10 µm) is observed when increasing the sonication time, promoting the creation of a more uniform nanoparticle dispersion. Therefore, the optimum sonication time was set at 3 h for the manufacturing of the nanocomposites. 

Figure 3 shows the SEM images of the fracture surfaces of the final nanocomposites. Here, it can be observed that the distribution of the MNPs is significantly affected by both the MNP content and the presence of expandable microparticles. In this regard, several facts can be noticed. On the one hand, there is a high sedimentation of the MNPs for each content. This sedimentation is especially prevalent in the case of the low MNP contents (Figure 3a–c). The reason for this lies in the fact that the viscosity of the mixture is lower in these cases, facilitating the sedimentation of the nanoparticles, according to Stoke’s law [37,38,39]. When increasing the MNP content, the viscosity of the mixture increases, and the sedimentation is less significant (Figure 3d,f). Here, the presence of the expandable microspheres promotes a drastic reduction in the sedimentation, inducing a far more homogeneous distribution of the MNPs inside the material (Figure 3e,g). This is explained by the fact that the expandable microspheres induce an increase in the viscosity of the mixture, thus, hindering the movement of the MNP aggregates. In addition, compared to the samples without MS, the presence of larger aggregates is slightly higher in this case due to the lower efficiency of the ultrasonication process as the viscosity increases.

Furthermore, the samples with 20 wt.% MNPs show a higher presence of larger aggregates (Figure 3f,g) in comparison to the rest of the samples due to the higher viscosity of the mixture. Here, it can also be noticed that the sample with MS also shows the presence of a certain generalized porosity, probably induced by the excessive viscosity of the mixture, that hinders the proper removal of entrapped air during degasification. 

### 3.2. Mechanical and Electromagnetic Heating Characterization of Nanocomposites

#### 3.2.1. Mechanical Analysis by DMTA 

Figure 4 summarizes the main results of the DMTA analysis of the nanocomposites. Here, it can be observed that the T_g_ decreases with the increasing amount of MNPs. This is an effect that has been observed previously in other systems, with the addition of different nanoparticles [40]. In this regard, the presence of the nanoparticles may hinder the chain mobility during curing, leading to a lower crosslinking degree, which would explain the slight detriment to the T_g_. In fact, several studies have shown a decrease in the T_g_ due to the addition of different micro- and nanoparticles through the previously described effect [40]. Furthermore, the presence of the expandable microspheres promotes a more drastic decreased in the T_g_. This can be explained by the size of the microspheres themselves, which may induce a higher hindering effect on the chain mobility during curing. In addition, the effect that the MS has on the dispersion of the MNP, which promotes a lower sedimentation and, thus, a higher homogenization of the distribution of the MNPs, leads to a more global hindering effect on the nanocomposite.

Furthermore, the effect on the storage modulus at room temperature is quite similar, with a slight reduction as the MNP content increases. The inclusion of the microspheres promotes a drastic reduction in the case of the 20 wt.% MNP samples. This reduction can be correlated, in addition to the higher hindering effect, as explained above, with a very prevalent porosity that affects the load transfer and, thus, promotes a drastic reduction in the storage modulus. 

However, except for the 20 MNP-15 MS samples, the variation in the storage modulus when compared to the reference one (neat epoxy) is not as prevalent, indicating that the addition of the MNP and MS does not have a very significant negative effect on the mechanical properties of the epoxy adhesive under room conditions.

#### 3.2.2. EM Heating Tests

Figure 5a shows the results of the EM heating test of the nanocomposites. Here, it can be observed that the temperature reached increases with the increasing content of the nanoparticles, as expected, due to the higher efficiency of the EM heating which, in turn, is due the higher number of electrical and magnetic pathways inside the material. Concerning the inclusion of the expandable microspheres, a slight increase in the heating efficiency is observed for the 14 wt.% MNP samples, whereas a similar behavior is observed for the 20 wt.% samples. This can be explained by the previously analyzed effect of the nanoparticle dispersion. In the case of the 14 wt.% MNP samples, the lower sedimentation of the nanoparticles, because of the inclusion of the microspheres, leads to a higher efficiency and homogenization of the resistive heating and, thus, the temperature reached is higher. However, in the case of the 20 wt.% samples, although the inclusion of the microspheres also leads to a lower sedimentation of the nanoparticles, increasing the heating efficiency, the higher generalized porosity due to the reduced removal of the entrapped air leads to a reduction in the heating efficiency. Therefore, the mentioned generalized porosity promotes the disruption of the electrical pathways. Therefore, the combination of both effects, which act in opposite ways, leads to a very similar EM heating efficiency, regardless the inclusion of the microspheres.

Furthermore, Figure 5b summarizes the detailed heating responses of the different samples, showing an example of the heating–cooling curves as well as several IR images during the EM heating test. It can be observed that the heating rate increases with the MNP content in the first stages of the transitory heating due to the higher efficiency of the resistive heating. In fact, it ranges from 23 to 104 °C/min for the 8 and 20 wt.% MNP samples, respectively. In this regard, it can be pointed out that the heating rate is relatively high, and the stable temperature is reached at 2–4 min, highlighting the high efficiency of EM heating in comparison to other conventional techniques. In this regard, Figure 5c summarizes the IR images of the samples during the stabilization phase of the EM heating.

In addition, the effect of the inclusion of the MS particles promotes a higher homogeneity of the heating, as expected, due to the more homogeneous distribution of the MNPs. This homogeneity is manifested in a less significant difference between the average and the maximum temperatures reached by the samples (4 °C vs. 70 °C in the case of 14MNP and 3 °C vs. 100 °C in the case of the 14MNP-15MS samples). Moreover, the heating rate is also increased with the addition of the MS particles from 60 to 80 °C/min (Figure 5d).

### 3.3. Proof-of-Concept Analysis of SLS Joints

In order to prove the applicability of the proposed nanocomposites as detachable adhesives, SLS tests were conducted with the simultaneous application of EM heating. Here, only the 14 and 20 wt.% MNP samples (with and without MS) were tested. The reason for this lies in the fact that the lower MNP contents (8, 10, and 12 wt.%) did not reach a temperature high enough to promote adhesive detachment. In this regard, it is important to reach at least the T_g_, as the mechanical properties are significantly reduced, a fact that is crucial for achieving the abovementioned purpose.

Therefore, Figure 6a shows the LSS values under the different conditions. It can be noticed that there is a clear decrease in the LSS with the increasing current, that is, the EM heating. Here, the inclusion of MS induces a reduction in the joint strength under every condition, probably due to the discontinuous effect that promotes the presence of these microspheres. More specifically, in the case of the 20 wt.% MNP samples, the decrease in the adhesive properties is much higher due to the prevalent porosity, as mentioned before. However, in terms of the LSS reduction under EM heating, there are no prevalent differences caused by adding MS. 

Furthermore, it can also be noticed that, at the intensity of 250 A, there is no significant variation in the LSS values, and there is even a slight increase when compared to room conditions, probably due to an error in the test itself. This lack of significant change is explained by the fact that, at his value of intensity, the temperature reached in the joint is not enough to promote a prevalent decrease in the mechanical properties. On the other hand, when increasing the intensity to 500 A, a prevalent decrease in the LSS value is observed. In every case, at 750 A, the reduction in the LSS value ranges from 60 to 80% from the room temperature value. Therefore, it can be proved that the method of EM heating is highly efficient for the development of detachable adhesives, as the adhesion strength is clearly reduced under the most favorable conditions of the EM heating.

Concerning the failure type, it can be observed that every case is dominated by adhesive failure (Figure 6b), with the presence of some local cohesive areas in the case of the lower MNP contents and lower intensity values (highlighted regions in Figure 6b). Here, the inclusion of MS does not seem to affect the failure behavior. This can be explained by the fact that, although the presence of the MS promotes a higher homogeneity of the nanoparticles, the inclusion of these microspheres can also act as a stress concentrator.

## 4. Conclusions

Detachable adhesives based on a Fe_3_O_4_-nanoparticle-doped epoxy resin were proposed. 

It was observed that the ultrasonication process is an effective way of achieving a homogeneous nanoparticle distribution. More specifically, it can be noticed that, by increasing the sonication time, there is a prevalent breakage of the larger aggregates, leading to a reduction in the average size of the agglomerates. Therefore, the optimum sonication time was set to 3 h so as to obtain a proper nanoparticle distribution.

On the other hand, the effect of the magnetic nanoparticle (MNP) content was also explored. It was elucidated that the MNP sedimentation is higher at lower MNP contents due to the lower viscosity of the mixture, according to Stoke’s law. In addition, the inclusion of expandable microspheres (MS) promotes a lower sedimentation due to the increasing viscosity but also leads to a higher porosity, especially in the case of very high MNP contents (20 wt.%).

The EM heating efficiency was also evaluated. We found that the temperature reached increases with the induction current and MNP content due to a higher number of nanoparticles, which act as individual heaters. More specifically, the temperature increase was around 60–90 °C for 14 and 20 wt.% MNP, enough to reach the T_g_ and, thus, to promote the adhesive detachment. The inclusion of MS led to a slight to moderate increase in the heating efficiency due to the previously mentioned higher homogeneity of the dispersion. 

Finally, a proof-of-concept of the joint detachment in a single-lap shear (SLS) test was carried out. It was observed that the lap shear strength (LSS) was reduced from 60 to 80% from the reference conditions at the highest induction current applied to the joint, proving the high applicability of the proposed EM heating method for easier adhesive detachment. 

## Figures and Tables

**Figure 1 nanomaterials-12-03913-f001:**
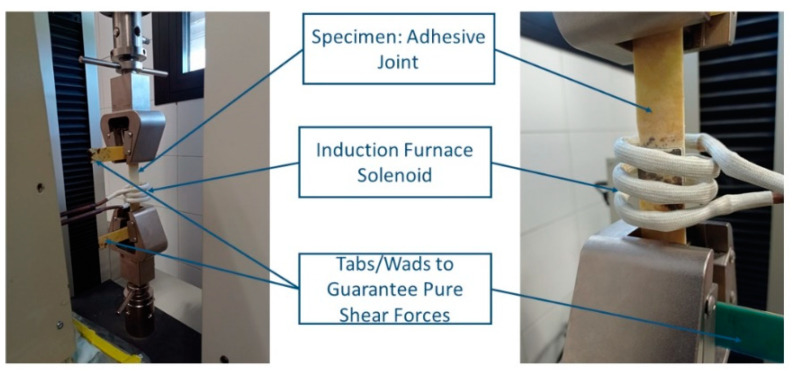
Experimental setup of single-lap shear test, where the adhesive joint is heated by induction.

**Figure 2 nanomaterials-12-03913-f002:**
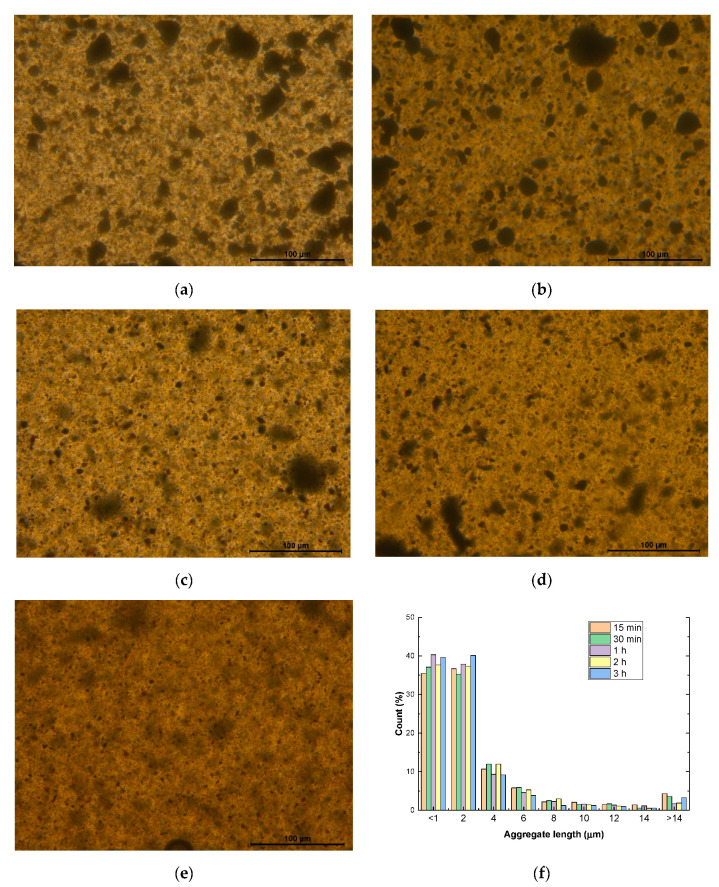
TOM images of the dispersion states after (**a**) 15, (**b**) 30 min, (**c**) 1, (**d**) 2, and (**e**) 3 h of sonication, and (**f**) histogram of the aggregate sizes for the different conditions.

**Figure 3 nanomaterials-12-03913-f003:**
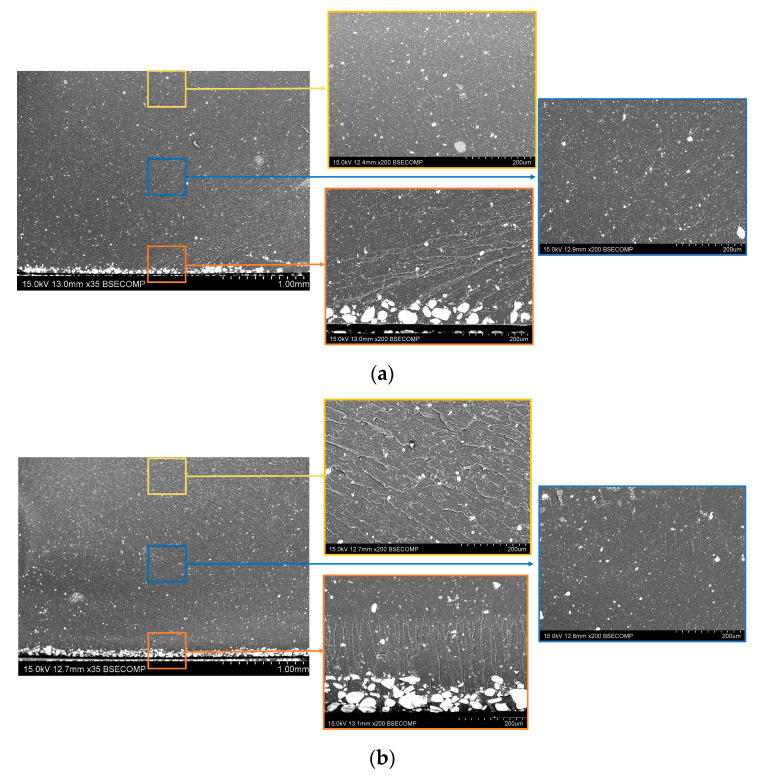
SEM images of the fracture surfaces of the (**a**) 8MNP, (**b**) 10 MNP, (**c**) 12MNP, (**d**) 14MNP, (**e**) 14MNP-15MS, (**f**) 20MNP and (**g**) 20MNP-15MS nanocomposites.

**Figure 4 nanomaterials-12-03913-f004:**
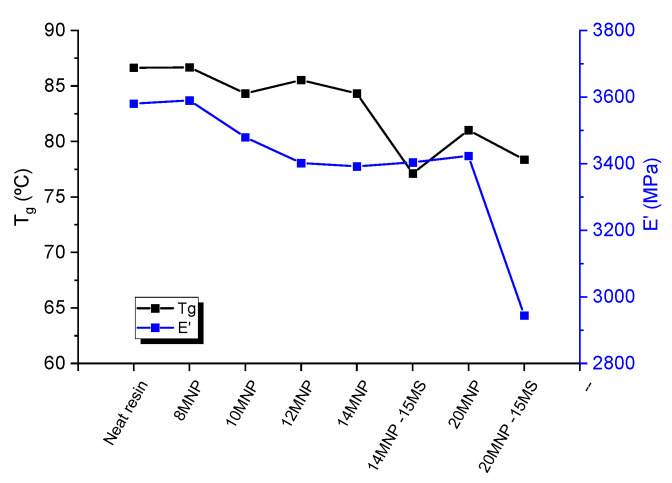
DMTA results showing the variation in the T_g_ and storage modulus under the different conditions tested.

**Figure 5 nanomaterials-12-03913-f005:**
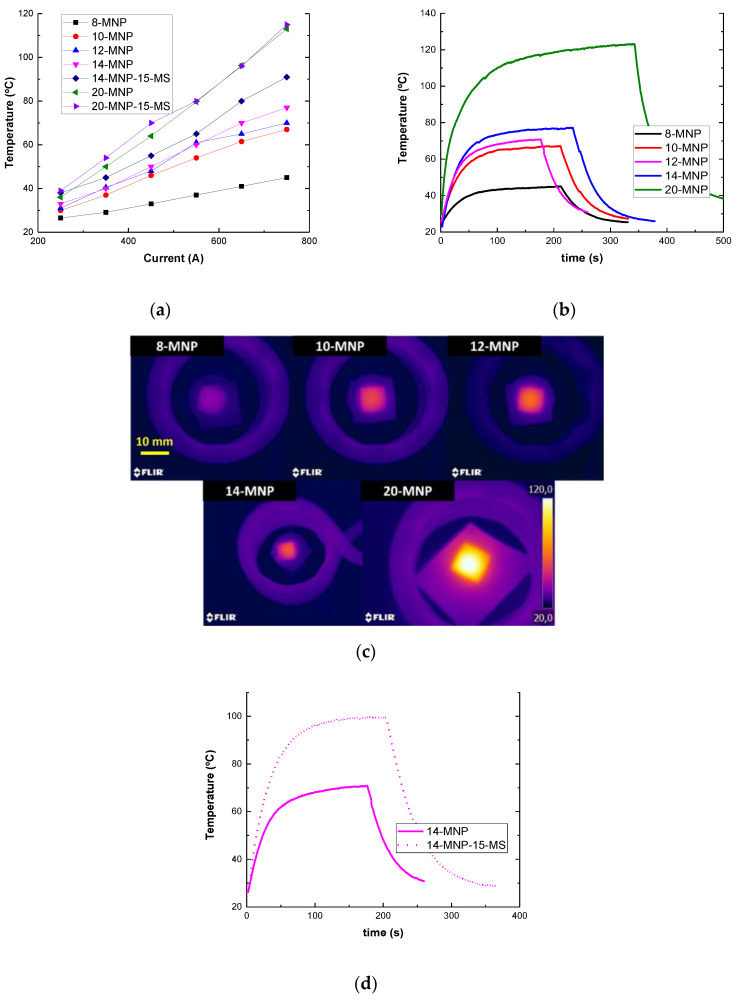
(**a**) Temperature reached for each condition as a function of the inductive current applied, (**b**) heating–cooling curves of EM heating as a function of the MNP content, (**c**) IR images at the stabilization phase, and (**d**) heating–cooling curves of the 14 wt.% MNP samples with and without MS inclusion.

**Figure 6 nanomaterials-12-03913-f006:**
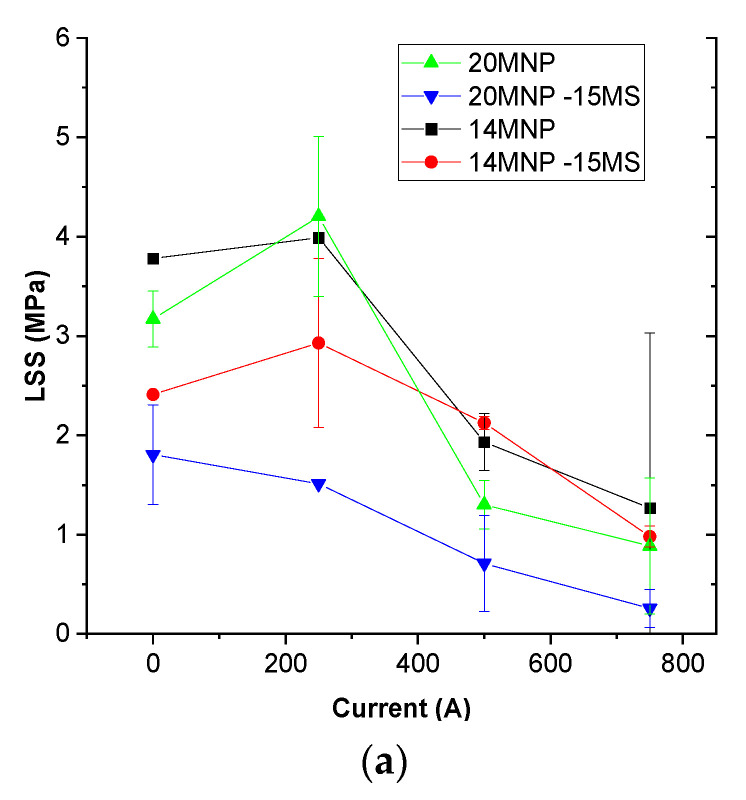
(**a**) LSS values and (**b**) images of the fracture surfaces showing the failure mode for the different conditions (the red circles denote the presence of cohesive failures).

**Table 1 nanomaterials-12-03913-t001:** Nomenclature and compositions (wt.%) of bulk samples manufactured.

Sample	wt.%		
	Epoxy Resin	MNP	MS
Neat resin	100	-	-
8MNP	92	8	-
10MNP	90	10	-
12MNP	88	12	-
14MNP	86	14	-
14MNP–15MS	71	14	15
20MNP	80	20	-
20MNP–15MS	65	20	15

## Data Availability

The data presented in this study are available on request from the corresponding author. The data are not publicly available due to technical limitations.

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
