# Peer review of "Fe_3_O_4_-Nanoparticle-Doped Epoxy Resin as a Detachable Adhesive by Electromagnetic Heating for GFRP Single-Lap Joints"

_nanomaterials, 2022, doi:10.3390/nano12213913_

Round 1

Reviewer 1 Report

General comments:

In this paper, a detachable adhesive based on a Fe3O4 nanoparticle (MNP) doped epoxy resin was proposed. The mechanical properties and heating capabilities are evaluated by an experimental approach. The authors proposed some scientific and new findings, which had certain academic value. However, given revisions are needed. The following are some comments:

Other comments:

Q1: The research background and gap is not clearly presented, the authors have to provide the specific contributions of each references rather than listing some of them together. In addition, some recent studies about adhesive technique should be added in order to enrich research background. For example, comparative study on joining quality of electromagnetic driven self-piecing riveting, adhesive and hybrid joints for Al/steel structure. Thin-Walled Structures 164 (2021) 107903; Mechanical properties and corrosion behavior of galvanized steel/Al dissimilar joints. Archives of Civil and Mechanical Engineering. 2021. 21(4) and so on.

Q2: For easier understanding of the paper, I suggest simplifying some long sentences. (Row 31-34, Row 78-82, Row 129-132, etc.)

Q3: The author please explain that how to control the thickness of nanocomposite adhesive at 1 mm?

Q4: There are suspected paragraphs in 2.1, please correct again. (Row 100-107)

Q5: Please check the unit in the text. Replace: "ºC" with: "℃". This is probably an error in the conversion to pdf format, but this should be corrected.

Q6: It is recommended to add a scale bar to the picture. (Figure 5 (c) and Figure 6 (b))

Reviewer 2 Report

The submitted manuscript is very interesting. Can be really disseminating for the readers. Each section is carefully and neatly written. No drawbacks or flaws can be found in it and it fullfils all the conditions of a good scientific paper.

I would love to accept ‘as is’, but a few tiny edits are necessary. Namely:

l. 45-46: stands ‘some of those...’ instead of ‘some of the…’

l. 100-107 Two paragraphs containing the template indications are left undeleted. Please remove.

l. 350: stands ‘his…’ instead of ‘this’

References are not formatted according to the journal style.

Round 2

Reviewer 1 Report

This manuscript has met the requirements for publication and is recommended for acceptance.